# Short-term effects of DAPAgliflozin on Lung fUNction, sleep apneas, and circulatinG surfactant protein B in Heart Failure with reduced ejection fraction (DAPA-LUNG-HF)

**DOI:** 10.3390/ijms26167696

**Published:** 2025-08-08

**Authors:** Massimo Mapelli, Irene Mattavelli, Elisabetta Salvioni, Cristina Banfi, Alice Mallia, Arianna Galotta, Valentina Mantegazza, Anna Garlaschè, Jeness Campodonico, Filippo Maria Rubbo, Chiara Paganin, Teresa Maria Capovilla, Rebecca Caputo, Mauro Contini, Paola Gugliandolo, Carlo Vignati, Beatrice Pezzuto, Giulia Grilli, Marco Scatigna, Alice Bonomi, Gianfranco Sinagra, Manuela Muratori, Piergiuseppe Agostoni

**Affiliations:** 1Centro Cardiologico Monzino, IRCCS, 20138 Milan, Italy; irene.mattavelli@ccfm.it (I.M.); elisabetta.salvioni@ccfm.it (E.S.); cristina.banfi@ccfm.it (C.B.); alice.mallia@ccfm.it (A.M.); arianna.galotta@ccfm.it (A.G.); valentina.mantegazza@ccfm.it (V.M.); anna.garlasche@ccfm.it (A.G.); jeness.campodonico@ccfm.it (J.C.); chiarap98@live.com (C.P.); rebecca.caputo@ccfm.it (R.C.); mauro.contini@ccfm.it (M.C.); paola.gugliandolo@ccfm.it (P.G.); carlo.vignati@ccfm.it (C.V.); beatrice.pezzuto@ccfm.it (B.P.); giulia.grilli92@gmail.com (G.G.); marco.scatigna@ccfm.it (M.S.); alice.bonomi@ccfm.it (A.B.); manuela.muratori@ccfm.it (M.M.); piergiuseppe.agostoni@ccfm.it (P.A.); 2Department of Clinical Sciences and Community Health, Cardiovascular Section, University of Milan, 20122 Milan, Italy; 3Cardiovascular Department, Azienda Sanitaria Universitaria Giuliano Isontina, 34128 Trieste, Italy; filippomariarubbo@gmail.com (F.M.R.); teresacapovilla@gmail.com (T.M.C.); gianfranco.sinagra@asugi.sanita.fvg.it (G.S.)

**Keywords:** dapagliflozin, SGLT2i, HFrEF, lung diffusion, surfactant binding proteins, sleep apneas

## Abstract

The mechanisms underlying the effects of dapagliflozin in heart failure with reduced ejection fraction (HFrEF) are not yet fully understood. This study aims to evaluate the effect of the drug on cardiorespiratory function by assessing alveolar–capillary membrane characteristics, sleep apnea, pulmonary and cardiac performance in stable HFrEF patients. Seventy-three patients with stable HFrEF were enrolled, with 66 completing the six-month follow-up. Analyses included assessment of the alveolar–capillary membrane by diffusion capacity, including its membrane diffusion and capillary volume components and measurements of proSP-B in the blood, an emerging biomarker of alveolar–capillary membrane function. Pulmonary function tests, overnight respiratory monitoring, and echocardiographic parameter collection were also conducted. After 6 months, a reduction in circulating proSP-B levels was observed (32.65 ± 13.36 at baseline vs. 30.86 ± 12.45 AU at 6 months, *p* for trend 0.0092), accompanied by improvements in echocardiographic parameters (left ventricle ejection fraction and pulmonary pressures). Pulmonary function tests and overnight respiratory monitoring showed no significant changes in lung diffusion, spirometry, or obstructive sleep apnea (apnea hypopnea index from 5.0 [1.1–16.6] at baseline to 6.2 [0.7–13.8]/h; *p* = n.s.). A significant reduction in central sleep apnea (CSA) was noted in the 13 patients with at least one CSA at baseline (15 [3–48] vs. 0 [0–18.5]/h, *p* = 0.017). Dapagliflozin demonstrates both hemodynamic and non-hemodynamic effects, particularly improving alveolar–capillary membrane function. This study highlights the multifactorial benefits of dapagliflozin in patients with stable HFrEF and the potential of proSP-B as a sensitive marker for evaluating therapeutic response.

## 1. Introduction

Recent clinical trials have led to the introduction of new medications for heart failure with reduced ejection fraction (HFrEF) treatment, which provide additional prognostic benefits [1,2,3,4]. Among these, sodium-glucose cotransporter 2 inhibitors (SGLT2i) have become a cornerstone of contemporary HFrEF treatment strategies [5,6]. While their positive effects on prognosis, cardiac remodeling, functional capacity, and natriuretic peptides are well-established [1,2,7], the mechanisms driving these benefits remain largely unclear. Indeed, both hemodynamic or non-hemodynamic pathways have been proposed, but none with convincing evidence [8,9,10].

To date, several potential targets of SGLT2i were poorly assessed as, for instance, the alveolar–capillary membrane function and sleep disturbances. The former is frequently impaired in HFrEF patients and can be assessed through lung diffusion capacity for carbon monoxide (DLCO) or by detecting the abnormal presence of surfactant proteins in peripheral blood [11,12,13]. In particular, the immature form of surfactant-derived protein type B (proSP-B) has emerged as a novel biomarker of alveolar–capillary membrane function and an indicator of overall HFrEF status [11,12]. Both DLCO and circulating proSP-B levels have been demonstrated to correlate with HF severity, to have a strong prognostic value, and have been shown to respond to specific acute and/or chronic treatments for HFrEF, such as inotropic infusions (e.g., levosimendan) or HFrEF drugs (e.g., sacubitril/valsartan) [11,14,15,16,17].

In parallel, emerging biomarkers such as soluble interleukin-1 receptor-like 1 (ST-2) have provided complementary prognostic insights, reflecting not only the hemodynamic state of HF patients but also their inflammatory and pro-fibrotic responses [18,19]. Furthermore, ventilatory abnormalities are commonly associated with an increased prevalence of sleep disorders in HF [20]. Sleep apnea is particularly prevalent among HFrEF patients, significantly contributing to patients’ poor quality of life, disease progression, and mortality [21,22,23,24,25]. Of note, central sleep apnea (CSA) is primarily linked to reduced cardiac output, while obstructive sleep apnea (OSA) is associated with intrathoracic fluid accumulation [26].

The aim of this study is to evaluate the effects of dapagliflozin—one of the two SGLT2i currently approved for HFrEF therapy—on circulating surfactant proteins, heart failure (HF) severity and fibrosis biomarkers, pulmonary and cardiac function, and sleep apnea in patients with HFrEF.

## 2. Results

Seventy-three patients were enrolled from January 2022 until July 2023. Seven patients (10%) were excluded from the final analysis because they interrupted the drug or the study (specifically: two subjects interrupted the study treatment for personal decisions; five individuals continued the study treatment but did not perform the follow-up evaluation), while none of the enrolled patients stopped the treatment for clinical reasons or drug-related complaints. All the remaining 66 HF patients (median age 67 [59–73] years) completed the 6 month (T2) evaluation, while 5 out of 66 did not perform the safety assessment at 2–4 weeks (T1). Table 1 reports the main baseline characteristics for the study population.

At 6 months, dapagliflozin confirmed its positive impact on echocardiographic parameters. Indeed, a significant reduction in left ventricular volumes and an increase in LVEF is observed at T2, along with a decrease in E/E′ and PAPs values (Table 2).

On average, at T0, FEV1, FVC, and DLCO were in the lower part of the normal range (Table 2). We did not observe a significant change in respiratory parameters, including FEV1 and FVC. As regards the alveolar–capillary membrane, we did not observe changes in DLCO and in its components Dm and Vcap. Differently, proSP-B showed a significant reduction over the course of the study (*p* for trend = 0.009) (Figure 1).

During the study period, the concomitant medical therapy was not significantly changed, the median dose of loop diuretic in patients taking furosemide at baseline (n = 35, 53%) remained stable (furosemide equivalent dose 25 [12.5–37.5] to 25 [25–25] mg/die, *p* = 0.952), and no new patients were started on diuretic. A significant increase in Hb values was observed (13.7 ± 1.6, 13.9 ± 1.4, 14.6 ± 1.7 g/dL at T0, T1 and T2, respectively, *p* for trend < 0.001). Differently, biomarkers such as NT-proBNP (from 852 [466–1858] at T0 to 944 [320–1854] pg/mL at T2, *p* for trend 0.051) and ST-2 (from 27.3 ± 6.8 to 28.1 ± 6.8 ng/mL, *p* for trend 0.354) did not significantly change during the study (Table 2). As expected, we observed a short-term worsening of creatinine (T1) with a complete recovery at T2 (*p* = 0.002 T0 vs. T1 with no significance for the other time-point comparisons) and no changes in potassium values (*p* for trend 0.088) (Table 2).

Nocturnal cardiorespiratory monitoring was assessed in 59 out of 66 patients and revealed minor, if any, sleeping characteristic abnormalities at T0. At T2, no effect on AHI, OSA, mixed apneas and hypopneas was observed, while a significant reduction in the number of CSA in the 13 (22%) patients with at least one CSA at baseline (n = 13; from 15 [3–48] at T0 to 0 [0–18.5] at T2, *p* = 0.017) was noted (Figure 2).

## 3. Discussion

The main and novel finding of this study is that, in our cohort of HFrEF patients, dapagliflozin therapy is associated with a significant reduction in circulant proSP-B albeit not paralleled by a concomitant improvement in DLCO (including its components Vcap and Dm) or pulmonary function tests. The reduction in proSP-B was observed shortly after the beginning of treatment and further continued thereafter. This protein, which is produced only by the alveolar cells, is usually elevated in the peripheral blood of HFrEF patients, correlates with DLCO, and undergoes a significant decrease under the influence of pharmacological or non-pharmacological interventions, such as inotropic infusions or the use of HF medications [16,17]. The lack of changes in DLCO and the lack of parallelism between proSP-B and DLCO after dapaglifozin therapy observed in the present study is not surprising for several reasons. First, we studied patients with a basically preserved DLCO. Second, we enrolled patients with HFrEF in a remarkably stable clinical condition when, differently from acute clinical settings, accumulation of fluids on the alveolar–capillary membrane is uncommon or minimal. Indeed, DLCO does not vary after ultrafiltration when performed in stable patients [27] while it improves in parallel with proSP-B after levosimendan treatment in acute HF [16]. As a matter of fact, DLCO in this HF setting is associated with a normal behavior or non-fluid dependent abnormalities of the alveolar–capillary membrane, albeit an increased intrathoracic fluid amount and lung congestion. Once again, this explains why ultrafiltration improves lung mechanics both at rest and during exercise without affecting DLCO.

The progressive reduction in proSP-B in the blood speaks, per se, in favour of an improved health status of the alveolar–capillary membrane. The mechanisms behind it are basically unknown, albeit a positive hemodynamic effect or more physiological activation of the various receptors and ionic pumps present on the alveolar–capillary membrane may both be suggested. Importantly, although the absolute reduction in proSP-B levels was modest (~6%), this finding is noteworthy given the highly selected and clinically stable nature of the study population, all of whom were on optimized guideline-directed medical therapy (e.g., 80% on sacubitril/valsartan, 95% on β-blockers, 83% on mineralocorticoid receptor antagonists). Similar modest but significant biomarker changes have been previously reported with other effective HF therapies, such as sacubitril/valsartan [17], suggesting that proSP-B may represent a sensitive marker of subclinical pulmonary vascular stress even in well-treated patients. As regards a possible antifibrotic effect of SGLT2i, we were unable to show any effect on the alveolar–capillary membrane, as membrane diffusion and fibrosis biomarkers were unchanged. However, the almost absent DLCO abnormalities at T0 and the relatively short observation period may have hampered our observation. As a matter of fact, subjects with a baseline-impaired DLCO (<80% pred at T0; n = 38) showed a trend in improvement at 6 months (*p* = 0.063).

Indeed, the hemodynamic impact of SGLT2i, although less pronounced compared to sacubitril/valsartan, has been previously documented. These drugs modestly lower blood pressure values [28] and reduce wedge pressure both at rest and during exercise [8]. According to our results, proSP-B could act as an early and sensitive marker, also compared to traditional HFrEF biomarkers related to congestion, like NT-proBNP. Indeed, in our population, which consists of well-treated, low-risk, stable HFrEF subjects (Table 1), dapagliflozin did not show a favorable effect on other parameters commonly linked to congestion, showing a neutral impact on DLCO, OSA, and standard spirometry. However, things may be different in patients with more advanced HF than those of the present study, as suggested by the favorable impact on CAS. Importantly, key echocardiography values, like E/E′, PAPs, LVEF and LV volumes, were significantly improved by the treatment. These changes seem to be independent of the well-known diuretic effect of the drug, as also shown by unchanged NT-proBNP values under a stable dose of loop diuretic.

The effect on sleep disturbances also deserves some comments. First, in the present cohort of HF patients, sleep abnormalities were minor and no significant changes were observed after 6 months of therapy with dapagliflozin in terms of AHI. Specifically, OSA did not decrease, and we were only able to identify a modest but significant reduction in CSA in patients who experienced it at baseline. However, these seemingly disappointing results are likely related to the low number of apneas observed in the present population at T0. In more advanced HFrEF populations, beneficial effects of SGLT2i therapy on AHI might become evident, as suggested in a previous trial by Jaffuel et al. [29] and recently demonstrated in a randomized trial [30].

### Limitations

This study has some limitations. Importantly, due to ethical reasons present since the publication of the most recent HF guidelines, it is not randomized, and therefore, a direct comparison of interventions is not possible. However, patients were enrolled only if clinically stable and on optimized HF therapy for at least 6 months, with no changes during the study period. This approach aimed to minimize confounding effects and better isolate the impact of dapagliflozin. Furthermore, the monocentric, small sample-size nature of the study does not allow an automatic generalization of the results to other populations. In addition, most of the patients were in NYHA class II; therefore, the effects of the drug in more severe HF populations deserve to be studied in dedicated trials. Moreover, patients with advanced chronic kidney disease (CKD) were not included in the present cohort. As such, no conclusions can be drawn regarding the potential effects of SGLT2i or other HF therapies in this subset, despite their increasing prevalence in clinical practice. In this regard, it would be of great interest to investigate the effects of these drugs in severe CKD (i.e., those with eGFR 15–30 mL/min/1.73 m^2^) patients in future studies, particularly with respect to biomarkers (starting from proSP-B) and functional parameters. Finally, we studied patients treated with dapagliflozin and no other SGLT2i drugs such as empaglifozin. We do not know whether similar results can be obtained with different molecules.

## 4. Materials and Methods

We enrolled a cohort of consecutive stable HFrEF patients who were referred to the HF Unit outpatient clinic of Centro Cardiologico Monzino, IRCCS in Milan. Inclusion criteria were age > 18 years, ability to sign study informed consent, stable clinical condition (i.e., no HF hospitalizations, devices implantation, surgical or percutaneous revascularization and/or major changes in the HF therapy within the previous 6 months), left ventricle ejection fraction (LVEF) ≤ 40% (echocardiography), diagnosis of HFrEF and eligibility for treatment with SGLT2i according to the most recent guidelines [31]. Exclusion criteria listed contraindications to SGLT2i, moderate-to-severe chronic obstructive pulmonary disease (COPD), or an estimated glomerular filtration rate (eGFR) < 30 mL/min/1.73 m^2^ according to Modified Diet in Renal Disease (MDRD) criteria [32]. Patients who met the study’s inclusion and exclusion criteria and signed the study agreement form underwent an initial evaluation (T0) that included standard spirometry and DLCO, collection of blood samples for proSP-B assay, transthoracic echocardiography, and nocturnal cardiorespiratory monitoring. Subsequently, patients started treatment with dapagliflozin 10 mg/day, as per the guidelines-directed HFrEF treatment. Between 2 and 4 weeks after starting the therapy (T1), a safety evaluation was performed, including blood sample collections and lung function tests. After 6 months of treatment (T2), patients underwent a reassessment of all parameters evaluated at T0.

### 4.1. Lung Function Tests

Standard spirometry and DLCO measurements were performed at rest according to the American Thoracic Society and the European Respiratory Society criteria (ATS/ERS guidelines) [33,34].

DLCO measurements were obtained while subjects were comfortably seated, with the single-breath constant expiratory flow technique (exhalation rate = 0.5 L/s) (Quark PFT Cosmed, Rome, Italy), DLCO measurements were corrected for hemoglobin (Hb) as previously described [35]. The diffusion subcomponents, membrane diffusion (Dm) and capillary volume (Vcap), were assessed by the Roughton and Forster method [36]. Alveolar volume was measured by CH_4_ decay slope during the single-breath constant expiratory flow measurement [37].

### 4.2. Venous Blood Sample Collection, Specimen Handling and Assay

Blood samples were drawn after assuring that any intense physical effort was avoided in the previous 3 h and after 5 min of rest in sitting position. ProSP-B determination was performed as follows: fresh blood (5 mL) was drawn into Vacutainer tubes containing citrate 0.129 mol/L as an anticoagulant. Plasma was immediately prepared by means of centrifugation at 1500× *g* for 10 min at 4 °C, divided into aliquots and frozen at −80 °C until assayed. Immature form of proSP-B was assayed as previously described [38]. Briefly, in order to precisely resolve low-molecular weight proteins, equal amounts of plasma proteins (50 µg) were separated by one-dimensional SDS-PAGE on 15% polyacrylamide gels using a Tris-Tricine buffer system in non-reducing conditions [39]. The protein concentration was evaluated by the method of Bradford [40]. Immunoblotting was performed using the primary antibody against SP-B (rabbit anti-human SP-B H300; Santa Cruz Biotechnology, Santa Cruz, CA, USA) diluted at 1∶200 in 5% (*w*/*v*) non-fat milk in TBS-T, followed by an incubation with a secondary goat anti-rabbit antibody conjugated to horseradish peroxidase (Bio-Rad, Milan, Italy). Following transfer, membranes were stained with MemCode reversible protein stain (Pierce Biotechnology, Cramlington, UK) according to the manufacturer’s instructions to ensure equivalent loading of protein. Immunoreactive bands ranging from 42 kDa to 17 kDa detected by ECL were quantified by densitometry of exposed film using image analysis software (QuantityOne version 4.5.2; Bio-Rad, Milan, Italy). For each subject, data are reported as the ratio of band volume, after local background subtraction, versus the volume of the total proteins loaded and stained with MemCode. The values were also normalized versus the band volume of pooled plasma, loaded as control on each gel, and they are expressed as arbitrary units (AU). Inter-assay coefficient of variation was 12.1 ± 2.9%.

### 4.3. Transthoracic Echocardiography

Echocardiography examinations were performed using a Philips ultrasound machine (Epiq CVx—Philips Medical Systems, Andover, MA, USA) equipped with an X5-1 probe. Complete standard 2DTTE analysis was performed. Left chambers’ volumes and LVEF were measured from 4-chamber and 2-chamber views using the biplane Simpson’s method [41]. All echocardiograms were performed by well-trained operators.

### 4.4. Nocturnal Cardiorespiratory Monitoring

Nocturnal cardiorespiratory monitoring was recorded by a SOMNOtouchTM RESP device (SOMNOmedics, Randersacker, Germany) before starting dapagliflozin and after six months of treatment. The SOMNOtouchTM RESP device is composed of a nasal cannula, a pulse oximeter, two respiratory sensors positioned at the level of the manubrium and abdomen, and three thoracic electrodes for ECG recording. Apnea was identified as a reduction in the amplitude of the respiratory flow signal, defined as a respiratory flow amplitude < 10% of the preceding baseline value for at least 10 s, while hypopnea was defined as a reduction in respiratory flow < 50% of the baseline for at least 10 s. Guidelines also recommend using oxygen desaturation > 3% as a criterion to detect hypopnea [42]. Apneas were considered of central origin (CSA) when the interruption in respiratory flow was associated with the absence of thoracic and abdominal respiratory effort, obstructive (OSA) if respiratory thoracic activity or abdominal activity were present during a cessation in respiratory flow, and mixed when an initially CSA turned into OSA in its final phase [26]. Apnea and hypopnea indexes (AHIs) were calculated as the number of apneas and hypopneas per hour of estimated or measured sleep time, respectively. The AHI is the sum of apneas and hypopneas per hour of sleep.

The present research protocol complies with the World Medical Association Declaration of Helsinki and was approved by the Centro Cardiologico Monzino Ethical Committee (R 11637-22 CCM 1756). Each subject provided written informed consent to the study. This study was registered on Clinicaltrials.gov with ID NCT05770167. Study data were collected and managed using REDCap electronic data capture tools hosted at Centro Cardiologico Monzino IRCCS [43,44]. REDCap (Research Electronic Data Capture) is a secure, web-based software platform.

### 4.5. Statistical Analysis

Continuous variables were described as mean ± standard deviation (SD) in the case of normal distribution, and as median and interquartile range (IQR) otherwise. Categorical variables were expressed as counts (percentages). For continuous variables, differences between T0 and T2 were assessed with a paired T-test or a Wilcoxon-signed-rank test, as appropriate. Missing data at time-point T1 were imputed using the PROC MI (multiple imputation) procedure with the Markov Chain Monte Carlo (MCMC) method. To analyze variables measured at the three protocol-specified time points (T0, T1, and T2), a repeated-measures ANOVA model was applied, after log-transformation of variables with a skewed distribution. A sensitivity analysis was performed by repeating the analyses in patients with complete data for all three time points. For pairwise comparisons, a Bonferroni correction was used to account for multiple testing. All tests were two-sided and a *p*-value < 0.05 was considered statistically significant. Statistical analyses were performed with SAS software, version 9.4 (SAS Institute, Cary, NC, USA) and SPSS Statistics V.29 (SPSS Inc., Chicago, IL, USA).

## 5. Conclusions

In conclusion, our trial conducted in a well-treated population of clinically stable subjects with moderate HFrEF, highlighted a beneficial impact of dapagliflozin on key alveolar–capillary unit parameters like proSP-B and CSA, despite a neutral effect on lung diffusion and pulmonary function tests. Taken together, these data help to understand the main mechanisms of action of this important therapy and, once again, highlight the potential of proSP-B as an early and sensitive marker for evaluating therapeutic response.

## Figures and Tables

**Figure 1 ijms-26-07696-f001:**
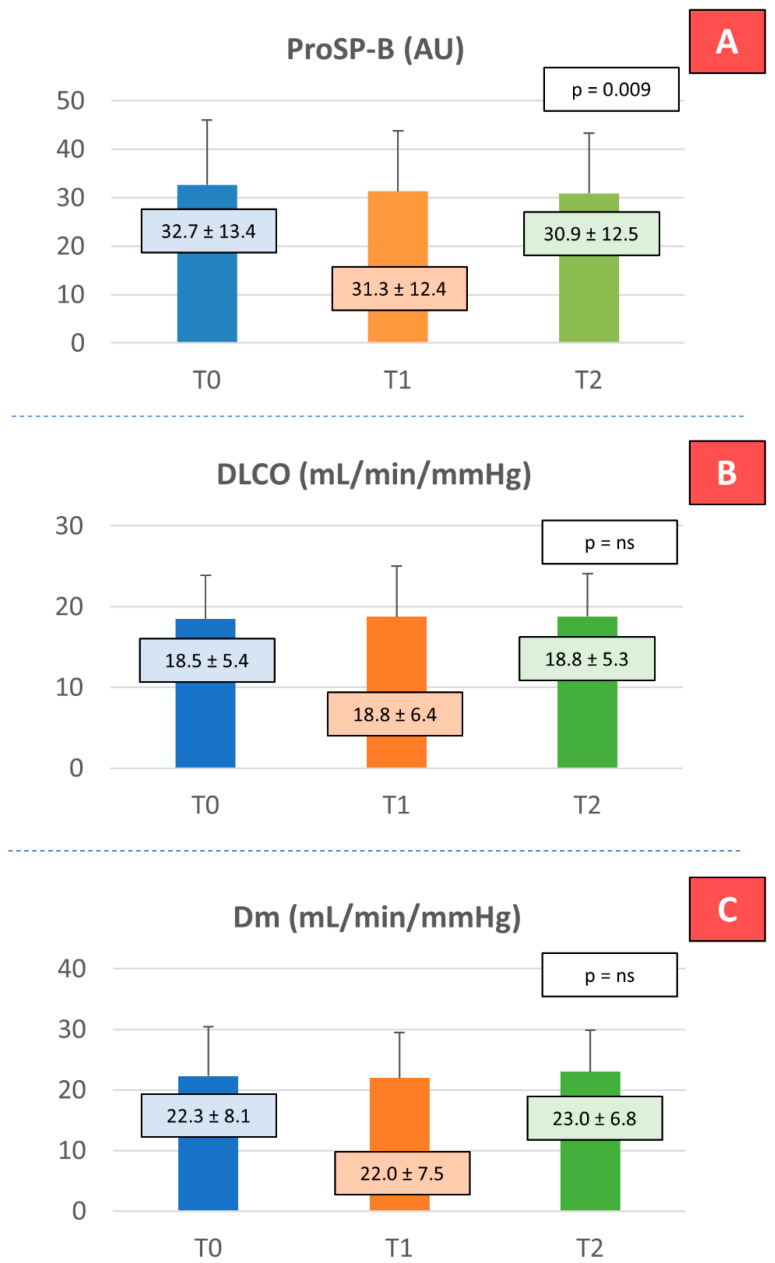
Effects of dapagliflozin on alveolar–capillary membrane function. (**A**) ProSP-B: surfactant-derived protein type B; (**B**) DLCO: lung diffusion capacity for carbon monoxide; (**C**) Dm: membrane diffusion; T0: initial evaluation; T1: 2–4 weeks after starting the therapy; T2: after 6 months of treatment. To analyze these variables a repeated-measures ANOVA model was applied and a *p*-value < 0.05 was considered statistically significant. n = 64 for proSP-B, n = 51 for DLCO and n = 47 for Dm.

**Figure 2 ijms-26-07696-f002:**
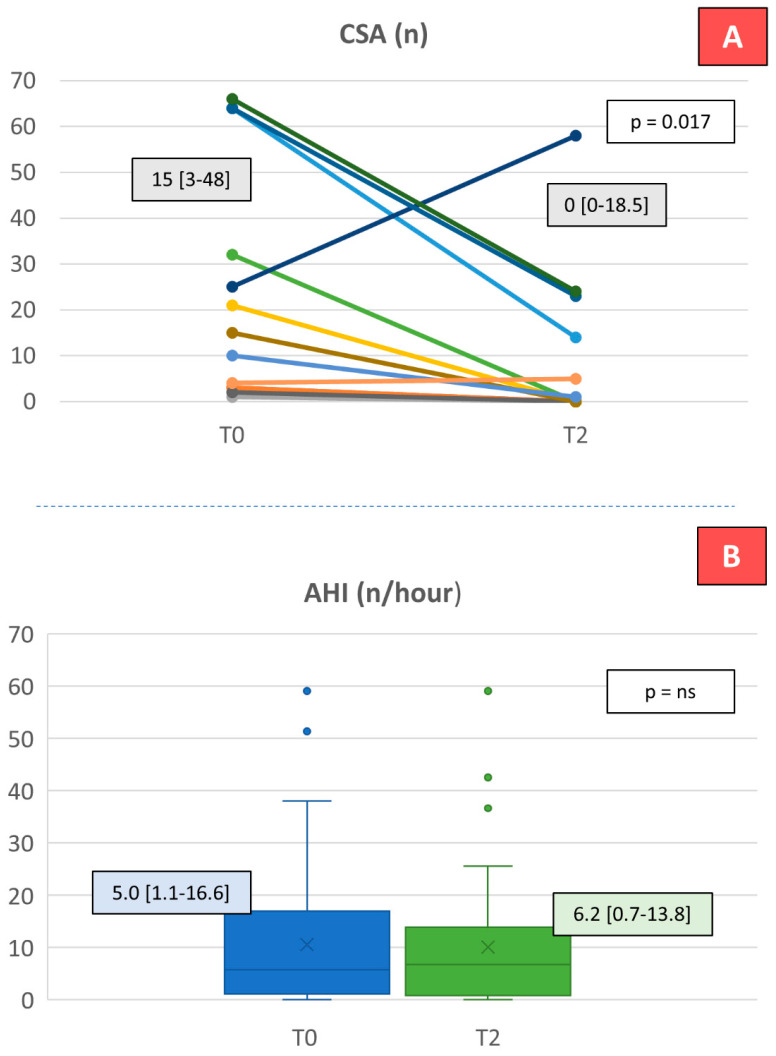
Dapagliflozin effects on sleep apnea (n = 59). (**A**) CSA: central sleep apnea; (**B**) AHI: apnea hypopnea index; T0: initial evaluation; T2: after 6 months of treatment. To analyze these variables, a Wilcoxon-signed-rank test was performed. Panel (**A**): analysis conducted only on patients with CSA > 0 at baseline (n = 13).

**Table 1 ijms-26-07696-t001:** Baseline characteristics of the population.

	Mean ± SD Median [IQR]
Males (n, %)	56 (85%)
Age (years)	66.5 [59.0–72.8]
BMI (kg/m^2^)	26.57 ± 3.51
Ischemic etiology (n, %)	33 (50%)
Hypertension (n, %)	33 (50%)
Type II diabetes (n, %)	7 (11%)
Dyslipidemia (n, %)	40 (61%)
Active smoker (n, %)	12 (18%)
Former smoker (n, %)	26 (39%)
COPD (n, %)	1 (2%)
Atrial fibrillation (n, %)	5 (8%)
ICD (n, %)	35 (53%)
CRT (n, %)	11 (17%)
ACEi (n, %)	6 (9%)
ARBs (n, %)	7 (11%)
Sacubitril/valsartan (n, %)	53 (80%)
β-blocker (n, %)	63 (95%)
MRA (n, %)	55 (83%)
Loop diuretic (n, %)	35 (53%)
Loop diuretic dose (mg/die) (n = 35)	25.0 [12.5–37.5]

Abbreviations: BMI: Body Mass Index; COPD: Chronic Obstructive Pulmonary Disease; ICD: Implantable Cardioverter Defibrillator; CRT: Cardiac Resynchronization Therapy; ACEi: Angiotensin Converting Enzyme Inhibitors; ARBs: Angiotensin Receptor Blockers; MRA: Mineralocorticoid Receptor Antagonist.

**Table 2 ijms-26-07696-t002:** Main changes in echocardiographic, pulmonary, sleep monitoring, and hematochimical values during the trial.

	N	T0	T1	T2	*p* TREND ANOVA	*p* Value Bonferroni Adjusted		
						T0 vs. T1	T0 vs. T2	T1 vs. T2
ProSP-B (AU)	64	32.65 ± 13.36	31.29 ± 12.43	30.86 ± 12.45	0.0092	0.0277	0.0209	1.0000
DLCO (mL/min/mmHg)	51	18.46 ± 5.4	18.75 ± 6.26	18.76 ± 5.3	0.4175	1.0000	1.0000	1.0000
DLCOHb (mL/min/mmHg)	51	18.65 ± 5.44	18.55 ± 5.58	18.85 ± 5.27	0.6241	1.0000	1.0000	1.0000
DLCO%	51	70 ± 16	69 ± 15	71 ± 15	0.6241	1.0000	1.0000	0.7748
Dm (mL/min/mmHg)	47	22.27 ± 8.11	21.97 ± 7.53	23.0 ± 6.82	0.1783	1.0000	1.0000	0.3640
FEV1 (L)	61	2.68 ± 0.69	2.7 ± 0.67	2.66 ± 0.64	0.5193	0.7952	1.0000	0.2881
FEV1%	61	87.02 ± 16.31	87.83 ± 15.99	86.9 ± 15.14	0.8521	0.6004	1.0000	0.7355
FCV (L)	61	3.36 ± 0.86	3.4 ± 0.84	3.39 ± 0.84	0.4112	0.5215	1.0000	1.0000
FVC%	61	84.3 ± 15.71	85.18 ± 14.72	85.22 ± 14.76	0.2642	0.8538	1.0000	1.0000
Hb (g/dL)	66	13.74 ± 1.55	13.93 ± 1.44	14.59 ± 1.7	<0.0001	0.1141	<0.0001	<0.0001
Creatinine (mg/dL) *	66	1.07 [0.91–1.33]	1.12 [0.94–1.39]	1.08 [0.89–1.36]	0.0123	0.0017	0.1352	0.9555
Potassium (mmol/L)	66	4.44 ± 0.39	4.52 ± 0.47	4.4 ± 0.37	0.7714	0.2833	1.0000	0.1098
NT-proBNP (ng/mL) *	64	852 [466–1858]	793 [298–1771]	944 [320–1854]	0.0717	0.0453	0.4037	1.0000
ST-2 (ng/mL)	61	27.28 ± 6.81	27.02 ± 6.48	28.08 ± 6.8	0.2628	1.0000	0.6841	0.5015
VA (L)	51	5.57 ± 1.32	5.57 ± 1.34	5.59 ± 1.32	0.9264	1.0000	1.0000	1.0000
Vcap (mL) *	43	98 [79–115]	101 [80–126]	94 [75–117]	0.7040	1.0000	1.0000	1.0000
EDV (mL)	66	186 [145–232]		177 [129–225]	<0.0001			
ESV (mL)	66	113 [87–163]		110 [76–145]	<0.0001			
LVEF (%)	66	34.60 ± 7.82		37.54 ± 9.20	<0.0001			
E/E′	60	8.5 [7.0–13.0]		8.0 [6.0–9.8]	0.0050			
PAPs (mmHg)	56	27 [24–29]		25 [23–28]	0.0460			
TAPSE (mm)	65	20.13 ± 5.72		19.34 ± 4.29	0.2010			
AHI (n)	59	5.0 [1.1–16.6]		6.2 [0.7–13.8]	0.3660			
Hypopneas (n)	49	8.0 [3.5–23.0]		10.5 [2.0–23.0]	0.8110			
CSA (n)	50	0.0 [0.0–2.3]		0.0 [0.0–1.0]	0.8090			

Abbreviation: ProSP-B: surfactant-derived protein type B; DLCO: Diffusing Capacity of the Lungs for Carbon monoxide; DLCOHb: DLCO corrected for heamoglobin level; DLCO%: DLCO expressed as percent of the predicted; Dm: Diffusing capacity of the alveolocapillary Membrane; FEV1: Forced Expiratory Volume in 1 s; FEV1%: FEV1 as percent of the predicted; FVC: Forced Vital Capacity; FVC%: FVC as percent of the predicted; Hb: Hemoglobin; NT-proBNP: N-terminal pro–B-type Natriuretic Peptide; ST-2: Soluble interleukin-1 receptor-like 1; Vcap: diffusing capacity for the pulmonary capillary blood volume; EDV: End-Diastolic Volume; ESV: End-Systolic Volume; LVEF: Left Ventricle Ejection Fraction; PAPs: Pulmonary Arterial Pressures; TAPSE: Tricuspid Annular Plane Systolic Excursion; AHI: The Apnea–Hypopnea Index; CSA: Central Sleep Apnea. * Results are presented for the log-transformed variables.

## Data Availability

Repository of raw data available after acceptance: www.zenodo.org.

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
