# Peer review of "Short-term effects of DAPAgliflozin on Lung fUNction, sleep apneas, and circulatinG surfactant protein B in Heart Failure with reduced ejection fraction (DAPA-LUNG-HF)"

_ijms, 2025, doi:10.3390/ijms26167696_

Round 1
Reviewer 1 Report
Comments and Suggestions for Authors
In the current study, Dr. Mapelli et al. investigated the short-term effects of dapagliflozin on lung function, sleep apnea, and surfactant protein B in heart failure subjects with reduced ejection fraction. The authors found a statistically significant decrease in the levels of circulating surfactant protein B after 6 months of treatment with dapagliflozin. However, changes are minimal, and the biological role of those changes is unclear. The study is missing appropriate controls and mechanistic insight. Novelty and significance are unclear. The study raises more questions than it answers.
Concerns:
- The study lacks a placebo control group. It is unclear whether the minimal changes found in proSP-B and echocardiography parameters are due to the effect of dapagliflozin or represent a response to optimal medical therapy shown in Table 1.
- Although the changes found in levels of proSP-B are statistically significant, they are minimal, and their clinical significance is unclear. The non-heart failure control group is needed to understand the significance of those changes.
- The authors’ statement that proSP-B could act as an even earlier and more sensitive marker, compared to traditional HFrEF biomarkers, is not supported by experimental data.
- The authors' statement that dapagliflozin improves alveolar-capillary membrane function, potentially through an antifibrotic mechanism, is not supported by any experimental evidence. In fact, both cardiac markers associated with fibrosis, NT-proBNP and ST-2, demonstrated an increase in median and mean values, respectively, after 6 months of dapagliflozin treatment.
- Previous studies, appropriately powered and controlled, have demonstrated the effect of dapagliflozin in preventing heart failure progression and reducing central sleep apnea (e.g., PMID: 36027570, PMID: 39023499, PMC10291304). The novelty and significance of findings in the current study are unclear.
Author Response
In the current study, Dr. Mapelli et al. investigated the short-term effects of dapagliflozin on lung function, sleep apnea, and surfactant protein B in heart failure subjects with reduced ejection fraction. The authors found a statistically significant decrease in the levels of circulating surfactant protein B after 6 months of treatment with dapagliflozin. However, changes are minimal, and the biological role of those changes is unclear. The study is missing appropriate controls and mechanistic insight. Novelty and significance are unclear. The study raises more questions than it answers.
We thank the reviewer for their time, thoughtful comments, and critical evaluation of our work. We appreciate their perspective and the opportunity to address the concerns raised, which we believe will help us improve the clarity and scientific value of our manuscript.
Concerns:
1. The study lacks a placebo control group. It is unclear whether the minimal changes found in proSP-B and echocardiography parameters are due to the effect of dapagliflozin or represent a response to optimal medical therapy shown in Table 1.
We thank the reviewer for this important point. Unfortunately, a randomized controlled design was not feasible in this study, as for ethical reasons we could not withhold SGLT2i therapy from eligible patients. Indeed, participants were enrolled precisely because they were suitable candidates for SGLT2i treatment according to current HF guidelines. We fully acknowledge that the prospective but non-randomized nature of our study represents a limitation, and this has been clearly stated in the Limitations section at the end of the Discussion.
That said, we believe it is unlikely that the observed (albeit modest) improvement in functional parameters and proSP-B levels was due to background medical therapy alone. Patients were enrolled only if they were clinically stable and on optimized HF treatment for at least 6 months prior to inclusion, with no changes in pharmacological therapy, no device implantation, and no revascularization procedures (this part has been added in the inclusion criteria section, as also asked by Reviewer2). Importantly, no modifications to HF therapy were made during the follow-up period. This allowed us to isolate, as much as possible, the effect of dapagliflozin within the limits of a non-randomized design.
Following your important comment, a sentence has been added in the limitation section:
“However, patients were enrolled only if clinically stable and on optimized HF therapy for at least 6 months, with no changes during the study period. This approach aimed to minimize confounding effects and better isolate the impact of dapagliflozin.”
2. Although the changes found in levels of proSP-B are statistically significant, they are minimal, and their clinical significance is unclear. The non-heart failure control group is needed to understand the significance of those changes.
We thank the reviewer for raising this relevant point. We fully agree that, while the reduction in circulating proSP-B levels observed in our study is statistically significant, the absolute change is modest, and its direct clinical significance remains to be better clarified. However, we would like to highlight several important considerations.
First, the patients included in this study represent a highly selected and clinically stable cohort with optimized medical therapy for HF (e.g., 80% on sacubitril/valsartan, 95% on β-blockers, 83% on MRAs). These were not advanced or decompensated cases, but rather patients in whom further hemodynamic or biomarker improvement is inherently difficult to achieve. In this context, even a ~6% reduction in proSP-B levels is notable and supports the idea of a biological signal of benefit. Similar small but significant improvements were observed in our previous prospective cohort studies (e.g., PMID: 40166601), where SGLT2i therapy led to measurable benefits in VE/VCOâ‚‚ slope, LVEF, hemoglobin levels, and a reduction in estimated 2-year mortality (MECKI score).
Second, the role of proSP-B as a dynamic biomarker in HF is still emerging. In a previous study from our group (PMID: 36186983), we observed that sacubitril/valsartan—despite its strong hemodynamic effects—also led to a modest but significant reduction in proSP-B levels. This supports the concept that large absolute changes in this biomarker are not to be expected, but rather that consistent downward trends in response to effective HF therapies may reinforce its relevance as a marker of subclinical pulmonary congestion and capillary stress.
We agree that inclusion of a non-HF control group would help contextualize these findings and further clarify the biological meaning of the observed changes. Unfortunately, this was beyond the scope of the current study, but we believe it remains an important area for future research.
Following your important suggestion we have expanded the discussion section
“Importantly, although the absolute reduction in proSP-B levels was modest (~6%), this finding is noteworthy given the highly selected and clinically stable nature of the study population, all of whom were on optimized guideline-directed medical therapy (e.g., 80% on sacubitril/valsartan, 95% on β-blockers, 83% on mineralocorticoid receptor antagonists). Similar modest but significant biomarker changes have been previously reported with other effective HF therapies, such as sacu-bitril/valsartan [17], suggesting that proSP-B may represent a sensitive marker of subclinical pulmonary vascular stress even in well-treated patients.”
3. The authors’ statement that proSP-B could act as an even earlier and more sensitive marker, compared to traditional HFrEF biomarkers, is not supported by experimental data.
Thank you. We have re-phrased the discussion following your valuable suggestion. Certainly, the trial was not designed to directly compare the sensitivity of different biomarkers. However, we believe it is relevant that, in this well-treated and stable HF population, we were able to observe a significant reduction in proSP-B levels, whereas NT-proBNP and ST2 remained unchanged. We thank you very much for this insightful comment.
4. The authors' statement that dapagliflozin improves alveolar-capillary membrane function, potentially through an antifibrotic mechanism, is not supported by any experimental evidence. In fact, both cardiac markers associated with fibrosis, NT-proBNP and ST-2, demonstrated an increase in median and mean values, respectively, after 6 months of dapagliflozin treatment.
Thank you for this comment. We totally agree that a clear antifibrotic mechanism is not fully demonstrated. As you stated no changes were observed at 6 months in NT-proBNP/ST-2 values. So that, we erase the sentence.
5. Previous studies, appropriately powered and controlled, have demonstrated the effect of dapagliflozin in preventing heart failure progression and reducing central sleep apnea (e.g., PMID: 36027570, PMID: 39023499, PMC10291304). The novelty and significance of findings in the current study are unclear
We thank the reviewer very much for this important comment. The first two cited studies (PMID: 36027570 and 39023499) are indeed pivotal and significant works focusing on cardiovascular outcomes and echocardiographic reverse remodeling. These data are fundamental and highly useful in clinical practice. However, our study aimed to highlight different parameters more related to cardiorespiratory pathophysiology and potential “ancillary” effects of the drugs.
Considering the novelty of these molecules—quite atypical in their evolutionary history, starting from diabetic populations up to their mechanisms of action—we believe that the effects on respiratory variables (such as alveolar-capillary diffusion) and their related biomarkers (proSP-B) could represent a new and stimulating finding to better understand the mechanisms underlying the prognostic improvement.
Regarding the last cited study (PMC10291304), it certainly represents an important but unique study (with limited literature on the topic). For this reason, it is already cited in our discussion. We consider that confirming, even with some differences, the findings of this small previous experience in the literature remains important and significant—especially since the study by Jaffuel et al. enrolled only 18 patients.

Reviewer 2 Report
Comments and Suggestions for Authors
Dear authors,
I have received for review the article you submitted, entitled Short-term effects of DAPAgliflozin on Lung fUNction, sleep apneas, and surfactant protein B in heart failure with reduced ejection fraction (DAPA-LUNG-HF).
I would like to express my appreciation for the efforts made by the team of authors throughout the study and in writing this manuscript.
The manuscript submitted for review discusses a topical issue in the context of SGLT2i treatment, namely their short-term effect on respiratory function. It looks at the impact of SGLT2i on sleep disorders, circulating proSP-B, and the evolution of hemodynamic parameters.
I appreciate the quality of the methodology, statistical analysis, discussion of the results, and well-structured conclusions.
Even though the improvement in hemodynamic parameters observed in the study group was expected, among the conclusions outlined by the authors, I would like to emphasize the potential future value of circulating proSP-B levels as an early marker of congestion.
Overall, I consider the manuscript to be of good interest for publication, providing robust data with possible implications for the future, even in spite of the limitations imposed by the lack of randomization and the small size of the study group.
However, I would like to highlight a few aspects that may need to be supplemented in order to better outline the study.
1. Line 79: I believe that the status of stable clinical condition should be defined in the text for better identification of target patients.
2. Line 83: Did you consider extending the study to include those with eGFR 15-30 ml/min/1.73 m2, given the promising benefits in patients with stage 4 CKD? Would more severe renal dysfunction have confused the results regarding the predictability of circulating proSP-B in cases of congestion?
I look forward to the final version of your article.
Best of luck in all your endeavors!
Author Response
Dear authors,
I have received for review the article you submitted, entitled Short-term effects of DAPAgliflozin on Lung fUNction, sleep apneas, and surfactant protein B in heart failure with reduced ejection fraction (DAPA-LUNG-HF).
I would like to express my appreciation for the efforts made by the team of authors throughout the study and in writing this manuscript.
The manuscript submitted for review discusses a topical issue in the context of SGLT2i treatment, namely their short-term effect on respiratory function. It looks at the impact of SGLT2i on sleep disorders, circulating proSP-B, and the evolution of hemodynamic parameters.
I appreciate the quality of the methodology, statistical analysis, discussion of the results, and well-structured conclusions.
Even though the improvement in hemodynamic parameters observed in the study group was expected, among the conclusions outlined by the authors, I would like to emphasize the potential future value of circulating proSP-B levels as an early marker of congestion.
Overall, I consider the manuscript to be of good interest for publication, providing robust data with possible implications for the future, even in spite of the limitations imposed by the lack of randomization and the small size of the study group.
Dear Reviewer#2, we would like to thank you for the enthusiastic feedback given to our study. You pointed out all the most important aspect related to our study. We have answered your comments (see below) to improve the quality of our manuscript. Thanks again for the time and the effort spent to review our work.
However, I would like to highlight a few aspects that may need to be supplemented in order to better outline the study.
1. Line 79: I believe that the status of stable clinical condition should be defined in the text for better identification of target patients.
Thank you for your comment. This has been now specified in the test
“no HF hospitalizations, devices implantation, surgical or percutaneous revascularization and/or major changes in the HF therapy within the previous 6 months”
2. Line 83: Did you consider extending the study to include those with eGFR 15-30 ml/min/1.73 m2, given the promising benefits in patients with stage 4 CKD? Would more severe renal dysfunction have confused the results regarding the predictability of circulating proSP-B in cases of congestion?
Thank you for this very important point. As the reviewer is well aware, patients with severe CKD are typically excluded from randomized controlled trials and are therefore underrepresented in the evidence base supporting most guideline-directed medical therapies for heart failure. However, they are increasingly prevalent in real-world clinical practice and HF cohorts. In this trial, given its prognostic but non-randomized design, we chose to include only clinically stable HF patients, as demonstrated by biomarker levels, LVEF, and NYHA class. This approach helped minimize confounding factors (e.g., HF hospitalizations during the study period, frequent changes in loop diuretic dosing) and reduced the risk of dropout.
We believe that HF therapies (including SGLT2 inhibitors) may have an important role in this subset of patients, and it is possible that the decongestive effects of these drugs—reflected by improvements in congestion biomarkers and potentially in sleep apnea—could be even more pronounced. However, as these patients were not included in the trial, we cannot draw any conclusions. In response to your valuable suggestion, a few lines have been added to the limitations section.
“Moreover, patients with advanced chronic kidney disease (CKD) were not included in the present cohort. As such, no conclusions can be drawn regarding the potential effects of SGLT2i or other HF therapies in this subset, despite their increasing prevalence in clinical practice. In this regard, it would be of great interest to investigate the effects of these drugs in severe CKD (i.e., those with eGFR 15-30 ml/min/1.73 m2) patients in future studies, particularly with respect to biomarkers (starting from proSP-B) and functional parameters”
I look forward to the final version of your article.
Best of luck in all your endeavors!
